# Culturally Aware Natural Language Inference

**Jing Huang, Diyi Yang**
Stanford University
Department of Computer Science
`{hij, diyiy}@cs.stanford.edu`

## Abstract

Humans produce and consume language in a particular cultural context, which includes knowledge about specific norms and practices. A listener's awareness of the cultural context is critical for interpreting the speaker's meaning. A simple expression like *"I didn't leave a tip"* implies a strong sense of dissatisfaction when tipping is assumed to be the norm. As NLP systems reach users from different cultures, achieving culturally aware language understanding becomes increasingly important. However, current research has focused on building cultural knowledge bases without studying how such knowledge leads to contextualized interpretations of texts. In this work, we operationalize cultural variations in language understanding through a natural language inference (NLI) task that surfaces cultural variations as label disagreement between annotators from different cultural groups. We introduce the first Culturally Aware Natural Language Inference (CALI) dataset with 2.7K premise-hypothesis pairs annotated by two cultural groups located in the U.S. and India. With CALI, we categorize how cultural norms affect language understanding and present an evaluation framework to assess at which levels large language models are culturally aware. Our dataset is available at `https://github.com/SALT-NLP/CulturallyAwareNLI`.

## 1 Introduction

Language, as a tool of social interaction, is used in a cultural context that involves specific norms and practices. Cultural norms are behavioral rules and conventions shared within specific groups, connecting cultural symbols and values (Hofstede et al., 2010). They provide contextual knowledge to interpret the meanings behind actions and words. For example, when tipping is customary, a simple expression like *"I didn't leave a tip"* implies a strong sense of dissatisfaction, while in a culture where tipping is optional, the implicature no longer holds.

Figure 1: An example of how cultural differences in tipping norms lead to different interpretations. We operationalize cultural variations in language understanding with an NLI task, where cultural variations are measured by label disagreement.

As NLP systems reach billions of users across cultural boundaries, building culturally aware language models is an emerging requirement (Hovy and Yang, 2021; Hershcovich et al., 2022). By the term "**culturally aware**", we specifically refer to three levels of awareness in language understanding: (1) having knowledge of specific cultural norms; (2) recognizing linguistic context that invokes cultural norms; (3) accommodating cultural variations by making a culture-specific inference. A lack of cultural awareness can result in models with poor accuracy and robustness (Liu et al., 2021), social biases and stereotypes (van Miltenburg, 2016; Rudinger et al., 2017; Arora et al., 2023), and discrimination against annotators (Miceli and Posada, 2022). To incorporate cultural factors in NLP systems, recent work has built knowledge bases of cultural symbols, practices (Yin et al., 2022; Nguyen et al., 2023; Fung et al., 2022; Ziems et al., 2023; CH-Wang et al., 2023), and values (Johnson et al., 2022; Cao et al., 2023; Arora et al., 2023), along with methods to probe or elicit such knowledge from LLMs.

In this work, we take a step towards culturally aware language understanding. Unlike previous work focusing on building knowledge bases of cul-

tural norms and practices—level (1), we focus on how people make different inferences based on cultural knowledge, i.e., level (2) and (3). We operationalize cultural variations in language understanding through an NLI task that measures cultural variations as label disagreement between annotators from different cultural groups. We introduce **CALI**, the first **C**ulturally **A**ware Natural **L**anguage **I**nference dataset[1], with 2.7K premise-hypotheses pairs. Each premise is centered around a type of normative behavior. We ask annotators from two different cultural groups, one based in the United States and the other based in India, to label the entailment relationships in the context of their culture. The label variations between the two groups capture the cultural variations in language understanding. With the new dataset, we categorize how cultural norms affect language inference (Section 4) and evaluate at which levels large language models are culturally aware (Section 5). We find despite LLMs having knowledge of many cultural norms, especially ones associated with the U.S. culture, they lack the ability to recognize such norms in the NLI task, i.e., level (2), and adjust inference based on cultural norms, i.e., level (3). Our work highlights that cultural factors contribute to label variations in NLI and cultural variations should be considered in language understanding tasks.

## 2 Related Work

### 2.1 Cultural Factors and Norms

Modeling cultural factors in language has received increasing attention in the NLP community (Hovy and Yang, 2021; Hershcovich et al., 2022). Culture serves as common ground in communication, motivating the line of work to build cultural knowledge bases and probe cultural knowledge in LLMs, which includes commonsense knowledge (Yin et al., 2022; Nguyen et al., 2023; Keleg and Magdy, 2023; Palta and Rudinger, 2023), norms (Fung et al., 2022; Ziems et al., 2023; CH-Wang et al., 2023), values (Johnson et al., 2022; Cao et al., 2023; Arora et al., 2023; Ramezani and Xu, 2023), and knowledge cross modalities (Liu et al., 2021).

More importantly, culture interacts with language, shaping what we convey and how we convey it. In cross-cultural communication, cultural differences cause misunderstandings of speakers' intentions (Thomas, 1983; Tannen, 1985; Wierzbicka,

1991). Recent work in NLP has studied differences in time expressions (Vilares and Gómez-Rodríguez, 2018; Shwartz, 2022), perspectives over news topics (Gutiérrez et al., 2016), pragmatic reference of nouns (Shaikh et al., 2023), culture-specific entities (Peskov et al., 2021; Yao et al., 2023), figurative language (Kabra et al., 2023; Liu et al., 2023b). Our work connects the two lines of research by investigating how cultural knowledge affects language understanding. Instead of studying a specific type of expression, we measure language understanding through a generic NLI task.

Our work is also related to reasoning about social norms. Social norm reasoning tasks involve social judgement (Forbes et al., 2020; Shen et al., 2022; Fung et al., 2022; Ziems et al., 2023), intents and consequences (Emelin et al., 2021; Rashkin et al., 2018), and reference resolution (Abrams and Scheutz, 2022). A recent line of work has focused on the situational and defeasible nature of social norm reasoning (Rudinger et al., 2020; Ziems et al., 2023; Pyatkin et al., 2023). Similar to these prior works, we emphasize the culture-contingent nature of normative reasoning.

### 2.2 Natural Language Inference

Natural language inference (NLI) is one of the most fundamental language understanding tasks (Bowman et al., 2015). The task is to determine whether the given hypothesis logically follows from the premise. Our work revisits NLI through a cultural lens. Cultural variations are first manifested in cross-lingual NLI (Conneau et al., 2018), motivated curating language-specific datasets (Artetxe et al., 2020; Hu et al., 2020). Our work focuses on the cross-cultural aspect. Unlike cross-lingual NLI captures culture on task-level, our task measures cultural variations on the instance level.

The cultural-specific inference is related to the discussion on speaker meaning and implicature (Manning, 2006; Pavlick and Kwiatkowski, 2019). Implicatures are inferences likely true in a given context. They are useful for commonsense reasoning and pragmatic inferences (Gordon et al., 2012; Rudinger et al., 2020), both involve cultural norms.

Lastly, cultural variations are a source of "human label variations" (Plank, 2022). Along with work on uncertainty (Pavlick and Kwiatkowski, 2019; Chen et al., 2020; Nie et al., 2020), ambiguity (Liu et al., 2023a), we challenge the assumption that there is a unique label per premise-hypothesis pair.

---

[1]We de-emphasize "natural" as the majority of the examples in the dataset are generated by large language models.

# 3 A Culturally Aware NLI Dataset

We introduce a new NLI dataset that measures cultural variations in language understanding through label variations of the NLI task. We choose NLI as a starting point, as it is one of the most fundamental and widely used natural language understanding tasks. The dataset generation framework is shown in Figure 2.

## 3.1 Task Definition

Given a cultural norm, we generate a narrative or an opinion related to the normative behavior as the premise and three statements about the goal or consequences of the behavior as hypotheses. The task is to infer the entailment relationship between the premise and the hypothesis under a particular cultural context.

Due to the culture-contingent nature of the inference, the entailment relationship is inherently an implicature that is always cancellable, i.e., true in some cultural context but false in others. Following a strict definition of entailment may lead to most examples being labeled as "neutral". Hence, in addition to an absolute entailment relationship, we also want to capture which hypothesis is more likely to be true, i.e., more plausible, in a given cultural context. We present multiple hypotheses per premise and measure the interpretation of a premise as a ranking of entailment relationships over the set of hypotheses.

## 3.2 Collecting Cultural Norms

A major challenge in measuring the influence of cultural norms on language understanding is to identify norms that have strong cultural variations. We survey the rich literature on cultural differences around social norms, especially the ones that study norm variations between the U.S. and India, such as tipping norms (Lynn and Lynn, 2004), dining etiquette (Hegde et al., 2018), wedding customs (Buckley, 2006), bargaining practices (Druckman et al., 1976), politeness (Valentine, 1996), etc. We also collect cultural norms from the Wikipedia corpus[2] and online forums where people from different cultures interact, such as the Subreddit r/AskAnAmerican[3]. Note that we only extract norms mentioned without directly including the post or comment from users in our dataset.

---

[2] https://huggingface.co/datasets/wikipedia
[3] https://www.reddit.com/r/AskAnAmerican/

Besides norms that are likely differ between the two cultures, we also sample a set of social norms from the NormBank (Ziems et al., 2023) that are likely consistent between the two cultures. This set allows us to measure label variations when cultural variations are not presented. We describe how to convert each norm into a premise in Section 3.4.

## 3.3 Representing Cultural Norms

Existing work on the computational representation of social norms converges on the rule of thumb (RoT). However, for cultural norms, RoT does not provide a structured way to specify the context where the norm applies. For example, having soup for dinner is normal, but it is whether one should have soup before or after the main dish that captures the cultural variation (Palta and Rudinger, 2023). Following earlier work on behavioral norms, we adopt the script representation (Schank and Abelson, 1977; Bicchieri, 2000), where behaviors conforming to a script form a causal chain to achieve a goal. The context of an action hence can be represented by structured elements of the script, such as the goal of the actor, previous or next actions, identities of the actors, objects involved, and changes in states. These structured elements capture fine-grained variations of norms, which allows us to generate premises and hypotheses targeting at these differences.

## 3.4 Model-in-the-loop Generation

With the script representation, we describe how to embed a cultural norm into a sentence context to generate a premise and hypothesis pair. The structured representation of cultural norm allows us to leverage LLMs to generate premises and hypotheses, a promising approach to construct NLI datasets (Liu et al., 2022; Chakrabarty et al., 2022). Specifically, we first prompt ChatGPT/GPT-3.5-turbo to generate premises and hypotheses from the script, followed by human editing on the generated content. We discuss the generation process in details below and list all prompts in Appendix A.1. For about 20% of norms, LLMs failed to generate suitable context. We discuss these cases in Appendix A.2. In these cases, We fall back to searching for sentences containing the behavior from existing corpus, mainly through Google Books Ngrams Viewer[4]. As a last resort, authors manually write about 5% premises and hypotheses.

---

[4] https://books.google.com/ngrams/

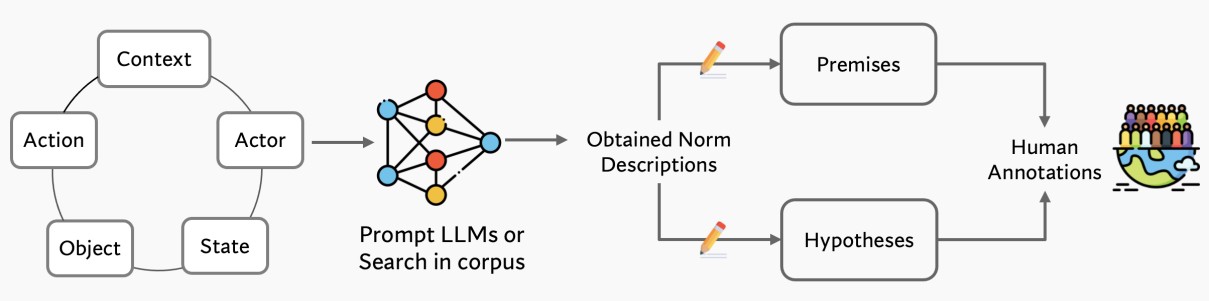

Figure 2: An overview of the dataset generation framework. Our framework takes in a normative behavior, represented as a behavioral script, and generates a premise that describes or comments on the behavior by prompting LLMs or searching in the existing corpus. Then, we prompt LLMs to generate hypotheses to infer different elements in the script. After light post-editing, we ask two cultural groups to annotate the entailment relationship between the premise and hypothesis where cultural differences surface as label variations.

**Premise Generation**  Given a behavior in the form of a verb phrase, we prompt LLMs to generate narratives or opinions related to the behavior, for example, "Describe a scene of tipping the waitress at a restaurant." In the post editing, we remove the target of the inference. For example, if the target of inference (hypothesis) is the cause of a reaction, we remove the previous action but keep the reaction. We also remove cultural indicators, e.g., names and countries, and irrelevant content.

**Hypothesis Generation**  Hypotheses ideally should cover a diverse set of interpretations. We encourage diversity by specifying different conditions in the prompt: 1) specifying different levels of certainty, such as "write statements that are likely true" or "write statements that are definitely false"; 2) specifying different inference targets, such as "what is the intention of the speaker" or "what are different interpretations of the reaction". We again prompt LLMs with a set of instructions and manually select three hypotheses per premise.

**Artifacts in Generated Examples**  LLMs generated contents may contain artifacts, societal biases, and toxic texts (Lucy and Bamman, 2021; Bender et al., 2021; Sheng et al., 2021; Yuan et al., 2021; Borchers et al., 2022). For our dateset, we observe that underspecified prompts with only the behavior information occasionally lead to contents with particular styles and gender/cultural stereotypes. See failure cases in Appendix A.2. We mitigate these biases in two ways (1) control the generation by specifying more content (e.g., behaviors, situations, reactions) and style (e.g., a writing style or the structure of a particular sentence); (2) authors manually review all generated contents and remove gender/cultural indicators (e.g., people's names, country names). Moreover, such artifacts

do not affect the entailment relationships. We also do not observe any toxic content.

**Generation ≠ Awareness**  It is tempting to think that if a model can generate the hypotheses, the same model can also solve the inference task. However, there is a key difference between the generation task and the NLI task – the generation process is agnostic of the actual human label. For example, a model generates a hypothesis with a prompt asking for highly plausible hypotheses, but human annotators may label it as contradicting. Moreover, cultural variations are also not involved in the generation process, i.e., the prompt does not specify that the hypothesis should be plausible for a particular cultural group.

### 3.5  Collecting Annotations

We collect human labels under two cultural contexts using MTurk. The annotation effort has received an IRB exemption. We use two MTurk crowdsource worker pools as **proxies** to two different cultural groups. To ensure intergroup cultural differences and worker availability, we choose one group to be workers based in the United States (US) and the other group to be workers based in India (IN) using the *Locale Qualification* function on MTurk. We recruit in total 125 US and 35 IN workers. We follow the standard NLI task instructions with three modifications: 1) Each annotator labels three hypotheses at once. Presenting multiple hypotheses encourages annotators to compare which hypothesis is more plausible. 2) Replacing the two extremes "definitely true/false" with "very likely true/false" for reasons discussed in Section 3.1. 3) Using a finer-grained five-scale rating. The full annotation template is in Appendix B.2. To reduce randomness, we collect at least 5 labels per premise-hypothesis pair. We also apply extensive

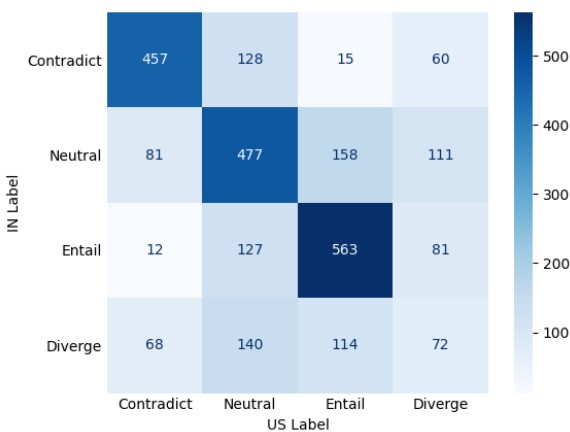

Figure 3: Distribution of premise-hypothesis pair labels in standard NLI label space, after label mapping, aggregating by majority vote, and filtering out "Diverge" examples. About 30% pairs, i.e., off-diagonal entries, are assigned different labels by the two cultural groups with at least one label being "Entail" or "Contradict".

quality control to make sure workers can achieve high accuracy on the standard NLI task. We describe the details in Appendix B.3.

## 3.6 CALI: Culturally Aware NLI Dataset

In total, we collect annotations for 900 premises and 2.7K hypotheses. We preserve labels from all annotators to reflect different cultural perspectives within a cultural group. However, to compare with existing NLI datasets, we map our 5-scale rating to the standard NLI label space and use majority vote for analysis. The mapping is determined by calibrating annotator labels against a set of 300 MultiNLI labels. Specifically, "1 (Very Likely False)" is mapped to "Contradict", "3 (Neutral)" is mapped to "Neutral", "5 (Very Likely True)" is mapped to "Entail". The mapping of "2" and "4" is annotator dependent, with the majority maps to "Neutral". Details of label mapping can be found in Appendix B.4. We then take the majority vote of the mapped labels. If a majority vote does not exist, we assign a label of `Diverge`. The label distribution is shown in Figure 3. To focus on cases where at least one cultural group reach a strong agreement, i.e., at least 75% annotators agreed on the same label, we discard 30% pairs and use the rest for analysis.

As culture-dependent inference is implicature by nature, annotator disagreement is expected to be more common than dataset focusing on textual entailment. Following the setup of SNLI (Bowman et al., 2015), we compute Fleiss kappa for a binary classification task per class with 5 randomly sampled raters per question over all examples passed

filtering. The Fleiss kappa for US, IN is 0.58, 0.51 for entailment, 0.37, 0.29 for neutral, and 0.58, 0.46 for contradiction. The level of agreement is lower than the range of 0.6-0.8 reported in SNLI, which uses a stricter sense of entailment and contradiction, but comparable with the range of 0.4-0.6 reported by Liu et al. (2023a) where ambiguity and implicatures are also involved.

## 4 Analyzing Cultural Variations

With CALI, we empirically investigate (1) how cultural knowledge leads to label variations in the NLI task and (2) what are other factors that contribute to label variations between the two cultural groups.

### 4.1 Categorizing How Cultural Norms Affect Language Understanding

Understanding how cultural knowledge leads to label variations in the NLI task is a missing link between having cultural knowledge and achieving culturally aware language understanding. To answer this question, we follow the categorization approach from the NLI literature, which is used to characterize lexical and world knowledge needed for recognizing textual entailment (Clark et al., 2007; Sammons et al., 2010; LoBue and Yates, 2011). For premise-hypothesis pairs in CALI that involve cultural norms and label variations, we manually annotate what linguistic phenomena and what types of cultural knowledge are involved in these examples. Examples are shown in Table 1.

**The Linguistic Dimension** We first consider what kinds of linguistic phenomena may invoke cultural knowledge. We divide the linguistic dimension into *semantic* and *pragmatic* categories. Semantic categories involve lexicon items with context-dependent meaning, including 1) lexical ambiguity due to multiple word senses, e.g., "bat" as "baseball bat" or "cricket bat' (Example #1 in Table 1); 2) context-dependent definitions, e.g., "formal dressing" as "wearing long dress"; 3) referring expressions, e.g., "it" in Example #2 and "she" in Example #8. Context-dependent expressions recruit common sense knowledge to fill in unspecified details. As a result, annotators with different cultural backgrounds resolve expressions to different entities, causing label variations.

Pragmatic categories are more common and complicated. We list two major subcategories: 1) conversational implicatures, including connotations of phrases describing normative or transgressive

| Premise and Hypotheses | Linguistic × Knowledge Category | Label Variation |
|---|---|---|
| **Example #1**
P: A boy hits a ball, with a bat, outside, while others in the background watch him.
H: The kid is playing in a baseball game.
(An example from SNLI) | **Linguistic**: Lexical ambiguity
**Knowledge**: Object; **US**: The most common bat-and-ball game is baseball. **IN**: The most common bat-and-ball game is cricket. | 55% E / N / C 50%

(SNLI: Entail) |
| **Example #2**
P: It's important to wash your hand and the spoon, as you eat the rice dish with it.
H: The last word "it" refers to your hand. | **Linguistic**: Referring expression
**Knowledge**: Object; **US**: It is common to eat rice dishes with utensils such as spoon. **IN**: It is common to eat rice dishes with hands. | E 60%
N
50% C |
| **Example #3**
P: The customer left a tip for the waiter at the restaurant.
H: The customer was satisfied with the excellent service. | **Linguistic**: Implicature
**Knowledge**: State; **US**: Tipping is customary. **IN**: Tipping is optional. | E 80%
60% N
C |
| **Example #4**
P: Nothing says "I trust this medication" like a commercial that lists off all the possible side effects.
H: Listing all possible side effects in a commercial is a sign of transparency and honesty that builds trust. | **Linguistic**: Flouting
**Knowledge**: Goal; **US**: It is normal to have advertising for medications with mandatory risk disclosure. | E
N 85%
55% C |
| **Example #5**
P: When asked if the violence issue is common, she replied "No, it's common."
H1: She agreed with the person who asked the question.
H2: She disagreed with the person who asked the question. | **Linguistic**: Politeness
**Knowledge**: Goal; **US**: "No" signals disagreement. **IN**: Avoid face threatening at the expense of contradictory statement. | H1 60%
N
40% H2 |
| **Example #6**
P: He had to admit that the house was taking shape. Most of the furniture was either hers or what they'd been given for their wedding.
H1: The couple asked their guests to buy furniture as gifts.
H2: The couple used monetary gifts from their wedding to purchase furniture. | **Linguistic**: Conversational implicature
**Knowledge**: Previous action; **US**: Couples communicate wedding gift preferences to guests through registry. **IN**: Money is the traditional wedding gift. | 85% H1
N
H2 80% |
| **Example #7**
P: "Did he leave a tip for the waiter at the restaurant?" His friends started laughing quietly as they asked.
H1: His friends did not know whether he left a tip or not.
H2: His friends thought he did not leave a tip for the waiter. | **Linguistic**: Conversational implicature
**Knowledge**: Previous action; **US**: Tipping is customary. **IN**: Tipping is optional. | H1
N
100% H2 60% |
| **Example #8**
P: Because Mona was the bride and Pia was her bridesmaid, she did not dress up in white for the wedding.
H1: Mona did not dress up in white for the wedding.
H2: Pia did not dress up in white for the wedding.
(An example adopted from WinoGrande) | **Linguistic**: Referring expression
**Knowledge**: Actor; **US**: White is the traditional color for wedding dresses. **IN**: It is okay to wear a white wedding dress, but traditionally bride wearing vibrant colors. | H1
N
70% H2 40% |

Table 1: Examples of cultural variations in the entailment label distribution between the two annotator groups **US** and **IN**. For entailment task, "P": Premise, "H": Hypothesis, "E": Entails, "N": Neutral, "C": Contradict. For plausibility task, "H1 (H2)": Hypothesis 1 (2), labels are "H1 (H2)": Hypothesis 1 (2) is more plausible and "N": Neutral, two hypotheses are equally plausible.

actions, e.g., "not leaving a tip" implies dissatisfaction (Example #3). Following the cooperative maxim, listeners are compelled to make pragmatic inferences by assuming speakers share the same set of norms. 2) violation of Gricean maxims or a clash with politeness, either due to flouting, e.g., Example #4 where the violation of quality can be interpreted as a touch of sarcasm or a strong assertion depending on the behavior and Example #5 where contradictory statements are used to avoid a face threatening situation (Valentine, 1996).

**The Knowledge Dimension** The knowledge dimension consists of the cultural norm used to generate each premise-hypothesis pair and the script

elements associated with the norm. These script elements provide definitions, e.g., "object" in Example #1 and #2, speaker intentions, e.g., "goal" in Example #4 and #5, causes, e.g., "state" in Example #3 and "previous action" in Example #6 and #7, and effects of actions, e.g., "actor" in Example #8.

## 4.2 Ambiguity and Subjectivity

We then examine other factors identified by previous research on label variations: ambiguity and subjectivity (Pavlick and Kwiatkowski, 2019; Zhang and de Marneffe, 2021; Jiang and Marneffe, 2022; Liu et al., 2023a).

**Ambiguity** To evaluate whether linguistic ambiguity alone can explain cultural variations observed, we annotate 50 premises from Ambient (Liu et al., 2023a) and compare with label variations observed on premises involved cultural norms. Unlike the context-dependent cases in Section 4.1, ambiguity in these examples cannot be resolved by culturally specific knowledge. We observed that only 10% of the premises have label variations, which is lower than cultural norm related premises. In other words, linguistic ambiguity does not fully account for the cultural variations observed in CALI.

**Annotator Subjectivity** Lastly, we examine whether annotators are simply inferring based on culture-specific knowledge, but ignore the linguistic context. We found annotators do assign non-neutral labels to hypotheses irrelevant to the premise when strong culture conventions are presented in the hypotheses only, however, aggregating by majority vote largely mitigates the subjectivity. We include these examples in Appendix C.

## 5 Evaluating Language Models

We first investigate how well models fine-tuned on existing NLI datasets perform on CALI. Specifically, whether entailment features learned from other datasets are transferable to CALI and whether these features are biased towards a particular cultural context. Then, we use the three-level cultural awareness defined in Section 1 and the categories developed in Section 4.1 as an evaluation framework to assess GPT-3.5/4 cultural awareness on NLI tasks, where we ask models to make culture-specific inference by prompting models with explicit cultural indicators.

| Model | All | US | IN |
|---|---|---|---|
| Entailment Classification Task – F1 macro | | | |
| S/MNLI | 0.70 | 0.71 | 0.70 |
| S/M/F/ANLI | 0.69 | 0.70 | 0.70 |
| WANLI | 0.69 | 0.68 | 0.67 |
| XNLI | 0.61 | 0.60 | 0.61 |
| X/ANLI | 0.66 | 0.67 | 0.65 |
| GPT-3.5 | 0.74 | 0.74 | 0.72 |
| GPT-3.5+US | 0.70 | 0.75 | 0.71 |
| GPT-3.5+IN | 0.70 | 0.68 | 0.73 |
| GPT-4 | **0.76** | **0.80** | **0.75** |
| GPT-4+US | 0.74 | **0.80** | 0.71 |
| GPT-4+IN | 0.71 | 0.74 | 0.72 |
| Choice of Plausible Alternatives Task – Accuracy | | | |
| GPT-3.5 | 54.45 | 57.59 | 58.63 |
| GPT-3.5+US | 48.69 | 52.36 | 53.93 |
| GPT-3.5+IN | 43.16 | 45.26 | 46.84 |
| GPT-4 | **58.12** | **67.54** | **59.16** |
| GPT-4+US | 56.54 | **67.54** | 58.64 |
| GPT-4+IN | 50.79 | 56.54 | 52.88 |

Table 2: F1 macro for entailment classification and accuracy for choice of plausible alternatives w.r.t. three sets of labels. Fine-tuned models are referred as the name of fine-tuning dataset.

## 5.1 Entailment Classification

We first evaluate models on two NLI tasks. For each task, we consider three sets of labels: "culturally generic" labels, i.e., "All", which is the majority vote from both groups and "culturally aware" labels, i.e., US (IN) labels, which is the majority vote from US (IN) annotators.

**Tasks** We cluster premise-hypothesis pairs by the normative behavior they covered and select the 40 largest clusters, which yields 500 pairs in total. The label distribution is roughly 25%, 50%, 25% for contradiction, neutral, entailment, with 30% labels differ between the two groups. For the entailment classification task, we perform a binary classification that treats only "entail" as positive. We do not use the 3-way classification setup, due to a distribution shift in labels discussed in Section 3.6. For each model, we take the probability of the "entail" class as the prediction and whether the majority vote from annotators is "entail" as the ground-truth.

The choice of the plausible alternative task takes in a premise and two hypotheses. We use a 3-way classification to predict which hypothesis is more likely to be true or if there is no clear preference between the two. The task is suitable for norm-based inference, as there are always exceptions to the norm that can cancel out the entailment.

**Models** We consider five state-of-the-art models fine-tuned on different NLI dataset (with ac-

curacy on the standard NLI test set – MultiNLI dev mismatched): SNLI + MultiNLI (90.49)[5], SNLI + MultiNLI + FEVER NLI + ANLI (90.6)[6], WaNLI (80.46)[7], MultiNLI + XNLI (87.58)[8], and MultiNLI + XNLI + ANLI (86.11)[9]. We also considered zero-shot prompting GPT-3.5/4 with two setup: one using NLI tasks prompt, the other adding locale information as culture indicators, e.g., "*Let's think as someone who lives in the United States.*" or "*Remind yourself of American culture*". We select prompts with the highest accuracy on a validation set (details in Appendix D.1).

Results are shown in Table 2. We observe that (1) models fine-tuned on existing NLI datasets have similar accuracy on all three sets of labels, i.e., not culturally aware but also not culturally biased. (2) GPT-3.5/4 in general outperform fine-tuned models, but do not show consistent improvements on cultural-specific labels when prompted with culture indicators. Among models fine-tuned on the existing NLI datasets, the two models with the highest accuracy on MultiNLI still have the highest F1 on our test sets, suggesting entailment features learned from the existing datasets are transferable to our dataset. Fine-tuning on cross-lingual NLI unfortunately does not help – X/ANLI model has the highest false positive rate, while XNLI has the highest false negative rate.

## 5.2 Diagnosing Cultural Awareness

We perform error analysis to investigate at which levels the LLMs lack cultural awareness. We focus on evaluating GPT-3.5/4, which allows us to specify the cultural context through prompting.

**Three-level Diagnostic Tasks** We decompose the culturally aware NLI into three diagnostic tasks: (1) Knowledge task: Whether errors are due to a lack of cultural norm knowledge; (2) Context task: Whether models fail to recognize contexts that invoke the norm; (3) Inference task: Whether models can reason about the effect of norms on entailment. Together, these tasks explore whether we can steer the model towards culture specific

---

[5] https://huggingface.co/cross-encoder/nli-deberta-v3-large
[6] https://huggingface.co/ynie/roberta-large-snli_mnli_fever_anli_R1_R2_R3-nli
[7] https://huggingface.co/alisawuffles/roberta-large-wanli
[8] https://huggingface.co/alan-turing-institute/mt5-large-finetuned-mnli-xtreme-xnli
[9] https://huggingface.co/vicgalle/xlm-roberta-large-xnli-anli

| Behavior | | Knowledge | | Con-text | Infer-ence |
| --- | --- | US | IN | | |
| Leave a tip | GPT-3.5 | 1.0 | 1.0 | 1.0 | 0.50 |
| | GPT-4 | 1.0 | 1.0 | 1.0 | 0.70 |
| Touch one's feet | GPT-3.5 | 1.0 | 1.0 | 0.6 | 0.69 |
| | GPT-4 | 1.0 | 1.0 | 1.0 | 0.61 |
| Have a gift registry | GPT-3.5 | 1.0 | 0.0 | 0.0 | 1.0 |
| | GPT-4 | 1.0 | 0.0 | 1.0 | 0.5 |
| Receive first paycheck | GPT-3.5 | 1.0 | 1.0 | 0.0 | 0.73 |
| | GPT-4 | 1.0 | 1.0 | 0.83 | 0.64 |
| Share food | GPT-3.5 | 1.0 | 1.0 | 0.86 | 0.91 |
| | GPT-4 | 1.0 | 1.0 | 1.0 | 0.82 |
| All 40 behaviors | GPT-3.5 | 0.93 | 0.79 | 0.62 | 0.68 |
| | GPT-4 | 0.98 | 0.93 | 0.88 | 0.71 |

Table 3: Results of the three cultural awareness diagnostic tasks. We report accuracy on the knowledge/inference task and recall on the context task. We show per-behavior accuracies for five behaviors and averaged accuracy over all 40 behaviors.

inference by specifying the cultural context.

We evaluate GPT-3.5/4 on the three tasks through prompting. The knowledge task consists of 40 questions on cultural norms, such as "What amount is considered normal when giving a tip in {country}?". We compare model outputs against sources of norms in Section 3.2 to determine the correctness of the answer. The context task asks models to list cultural norms involved in a premise. We use recall as the metric to check whether models can retrieve the norm used to generate the premise. Lastly, the inference task is formulated as defeasible inference (Rudinger et al., 2020), where the norm is present as an update to either strengthen or weaken the entailment (details in Appendix D.2).

The results are shown in Table 3. On the knowledge level, there is indeed an accuracy gap between US and IN. However, this knowledge gap is reduced from 14% to 5% when switching from GPT-3.5 to GPT-4, leading to over 90% accuracy for both cultural groups. These results suggest that GPT-3.5/4 have the knowledge of most cultural norm tested, in the sense that the models can pass at least one set of behavioral tests in a QA format that requires such knowledge. With the improvement in cultural norm knowledge task accuracy, we also observe a 30% improvement in recognizing norm in the sentence context. Lastly, for the inference task, we find that even when models are provided with cultural norm knowledge, there is still a 30% error rate in updating the inference with the cultural norms.

| Category | GPT-4 US | GPT-4 IN |
|---|---|---|
| Lexical Ambiguity (7%) | 1.00 | 1.00 |
| Referring Expressions (5%) | 0.73 | 0.61 |
| Violation of Maxims (9%) | 0.74 | 0.82 |
| Other Implicatures (79%) | 0.81 | 0.72 |

Table 4: A breakdown of F1 macro for entailment classification per linguistic dimensions.

**Which linguistic phenomena make cultural norms hard to recognize?** Now we take a closer look at the gap between the knowledge task and the inference task, using the linguistic dimensions laid out in Section 4.1. We evaluate on the same set of examples used in Table 2, which has about 12% semantic and 88% pragmatic cases. We focus our analysis on the GPT-4 results with the culture indicator prompting.

Results are shown in Table 4. GPT-4 performs the best on the lexical ambiguity category and performs the worst on the Winograd schema challenge-style reference resolution task. For the reference resolution task, despite having the necessary cultural knowledge (as shown in the knowledge probing task), GPT-4 is less sensitive to cultural indicators presented in the prompt, but biases towards using syntactic cues. These results highlight that the challenges of culturally aware language understanding lie in not only the knowledge level but also the interactions between knowledge and language.

## 6 Conclusion

We present a framework to study cultural awareness in natural language understanding. We define cultural awareness at three levels: (1) having knowledge of cultural norms, (2) recognizing cultural norms in context, and (3) making cultural-specific inferences. Our work focuses on level (2) and (3). We operationalize cultural variations in language understanding through an NLI task and contribute the first culturally aware NLI dataset, CALI. With our dataset, we categorize how cultural norms influence language understanding and present an evaluation framework to assess at which levels LLMs are culturally aware. We show despite having knowledge of cultural norms, LLMs still lack awareness on levels (2) and (3). Our work highlights that cultural factors contribute to label variations in NLI and cultural variations should be considered in language understanding tasks. We advocate for the community to incorporate cultural awareness in NLP systems in future research.

## Limitations

Our work only studies two cultural groups. While we have already observed evidence that cultural variations contribute to NLI label variations, expanding the study to multiple cultural groups would be highly valuable. We expect the proposed model-in-the-loop data collection framework to be helpful for studying NLI label variations in other cultures.

We use two locale-based crowdsource work pools as proxies for two different cultural groups. However, culture can vary greatly within a locale due to social class, age, and gender (Tannen, 1985). With label aggregation such as taking the majority vote, norms of larger or privileged groups might be overrepresented. Hence, we do not claim that particular cultural variations identified in our dataset are generalizable to different populations within the United States and India. The collected dataset is best used for analyzing how cultural norms interact with language understanding, yet it can be limited as a knowledge base of cultural norms in the United States and India. With our new framework to study cultural variations in language understanding, we hope that future work can investigate finer-grained cultural variations.

## Ethics Statement

**Using LLMs for Data Generation** To generate premises and hypotheses at scale, we use a model-in-the-loop paradigm for data collection. Model-generated premises and hypotheses may contain artifacts and stereotyped gender or culture associations (Lucy and Bamman, 2021; Bender et al., 2021; Yuan et al., 2021). We discuss these artifacts and mitigation strategies in Section 3.4 and Appendix A.2.

**Annotation Considerations** This work has received an IRB exemption at the authors' institution. For annotator compensation, we aimed for a pay rate of at least $12/hour, which is compliant with the federal minimum wage in the U.S. and India.

## Acknowledgements

We would like to thank the anonymous reviewers for their valuable feedback. We are also grateful of members of SALT lab, especially Raj Sanjay Shah for help on dataset collection, Caleb Ziems, Camille Harris, Ella Li, Myra Cheng, Will Held, and Weicheng Ma for feedback on the work. This

work was partially sponsored by the Defense Advanced Research Project Agency (DARPA) grant HR00112290103/HR0011260656, and NSF grant IIS-2247357 and IIS-2308994.

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

## Supplementary Materials

## A  Dataset Generation Details

### A.1  Prompts Used for Premise and Hypothesis Generation

We list the full set of prompt templates that we have experimented with in Figure 4. For premises, given a normative behavior, we randomly sample two premise prompt templates. For hypotheses, we randomly sample one of the "goal" prompt and two others prompts. For prompt selection, we observe that prompts with more information specified generally produce higher quality outputs.

---

Prompts for premise generation
```
In 1-2 sentences, describe a scene of {behavior} at {location}.
In a sentence, write {sentence_type} of {behavior} at {location}.
Write {sentence_type} implying {behavior} at {location} is {judgement}.
In a sentence, write a narrative or a comment implying {behavior} at {location} is {judgement}.
Write a sentence with {style} implying {behavior} at {location} is {judgement}.
```

The value of each placeholder is as following
`{sentence_type}`: One of "a sentence", "a short sentence", "a narrative", "a comment", "a sarcastic comment".
`{behavior}`: A verb phrase describing an action, such as "leaving a tip".
`{location}`: A noun phrase describing a place, such as "restaurant".
`{judgement}`: One of "normal", "expected", "unexpected", "taboo". Norms sampled from NormBank come with this annotation, however, for data generation, we simply random sample two opposite judgement per behavior.
`{style}`: One of writing styles, such as "colloquial style", "Mark Twain style", "Ernest Hemingway style", and "Reddit style" or an example sentence, such as "the style and structure of 'I thought working on Liddy's campaign would be better than working on Bob's." where the example sentence is taken from an existing NLI dataset. We use style to improve the diversity of generated text, in particular, diversify sentence structures and choice of words.

Prompts for hypothesis generation, listed by inference type.
**Actor:** `In a single sentence, infer what is the relation between the speakers/people mentioned in the scenario.`
`Scenario: {premise}`
**Actor:** `Write three sentences with different hypotheses on the identity of people mentioned in the scenario.`
`Scenario: {premise}`
**Goal:** `Write three sentences with different interpretations of the intention of the person or the speaker in the following scenario.`
`Scenario: {premise}`
**Goal:** `Write three sentences with different interpretations of the reactions in the following scenario?`
`Scenario: {premise}`
**Previous Action:** `Given the following scenario, can you infer what was the person doing or what happens to the person before this?`
`Scenario: {premise}`
**Next Action:** `Given the following scenario, can you infer what would the person do after this?`
`Scenario: {premise}`
**Object:** `Given the following scenario, can you infer what objects have been involved?`
`Scenario: {premise}`
**State:** `Write a single sentence inferring what the person may feel in the following scenario: {premise}`
**Entailment:** `Write a few statements that are definitely true given the following premise: {premise}`
**Neutral:** `Write three sentences with different implications of the following premise: {premise}`
**Paraphrase:** `Write three paraphrases of the sentence with different interpretations: {premise}`

---

Figure 4: Prompts used for premise and hypothesis generation.

### A.2  Premise Generation Failure Cases

In general, we do not observe that ChatGPT/GPT-3.5-turbo generate toxic content on the prompts used, however, we observe degraded generation quality when given social norms are non-stereotypical. These cases can be viewed as "out-of-distribution" inputs for the model. We list four major failure cases below.

**Repeating content in the prompt**  Generated texts are simply repeats of the behavior and judgement specified in the prompt, without any new information.

```
Prompt: In 1-2 sentences, describe a scene of saying "please" and "thank you" at a restaurant when
doing so is unexpected.
Output: A customer at a busy restaurant unexpectedly says "please" and "thank you" to the waitstaff,
causing them to pause and smile with appreciation.

Prompt: In 1-2 sentences, describe a scene of calling strangers "Uncle" and "Aunty".
Output: In some cultures, it is customary to show respect by addressing strangers who are older as
"Uncle" or "Aunty," even if they are not.
```

**Wrong cultural associations**   Generated texts contain normative behaviors associated with the wrong culture. As we remove the culture indicator in the post-editing phase, this type of failure is less of a concern for the data generation process. However, this type of failure shows the need to improve cultural awareness in LLMs.

```
Prompt: In 1-2 sentences, describe a scene of greeting with three kisses on alternating cheeks at
airport when doing so is unexpected.
Output: As she stepped off the plane, her American friends greeted her with three kisses on
alternating cheeks, a customary greeting in their culture, but unexpected for her.
```

**Stereotyped responses**   Generated texts contain stereotyped responses, include gender and social role associations. For stereotyped gender association, we try to rewrite the premise with gender-neutral pronouns in the post-editing stage if possible. For stereotyped social role, we discard the generated output, as the stereotyped associations likely have cultural variations.

```
Prompt: In 1-2 sentences, describe a scene of knitting a scarf at bank.
Output: A woman sits in the waiting area of the bank, calmly knitting a scarf with a ball of yarn
and knitting needles in her lap, while waiting for her number.

Prompt: In 1-2 sentences, describe a scene of wearing a red dress on a wedding.
Output: The bride wore a stunning white gown while her bridesmaids wore matching red dresses,
standing out against the greenery of the outdoor wedding ceremony.
```

**Refuse to generate**   ChatGPT/GPT-3.5-turbo refuses to generate texts for certain norms, despite such norms are commonly practiced in some countries.

```
Prompt: In a sentence, write a narrative or a comment implying threatening to leave without a raise
at work is normal.
Output: I'm sorry, as an intelligent assistant, I cannot provide a statement that implies threatening
to leave without a raise at work is normal or acceptable behavior.
```

## B   Annotation Details

### B.1   Annotator Compensation

We aimed for a pay rate of at least $12 per hour. Workers were paid $0.1 for each example (50 words) annotated.

### B.2   Annotation Template

We show the annotation template for qualification and actual task in Figure 5a and Figure 5b.

### B.3   Annotation Quality Control

We apply extensive quality control to make sure the annotations are valid responses from annotators who can follow the NLI task instruction. First, annotators were required to have a HIT approval rate of 95%, a total of 50 approved HITs to participate in the research. We also set up a qualification test with four standard NLI questions and two locale-specific commonsense questions. About 50% of Amazon MTurk workers who have completed the qualification test passed the test.

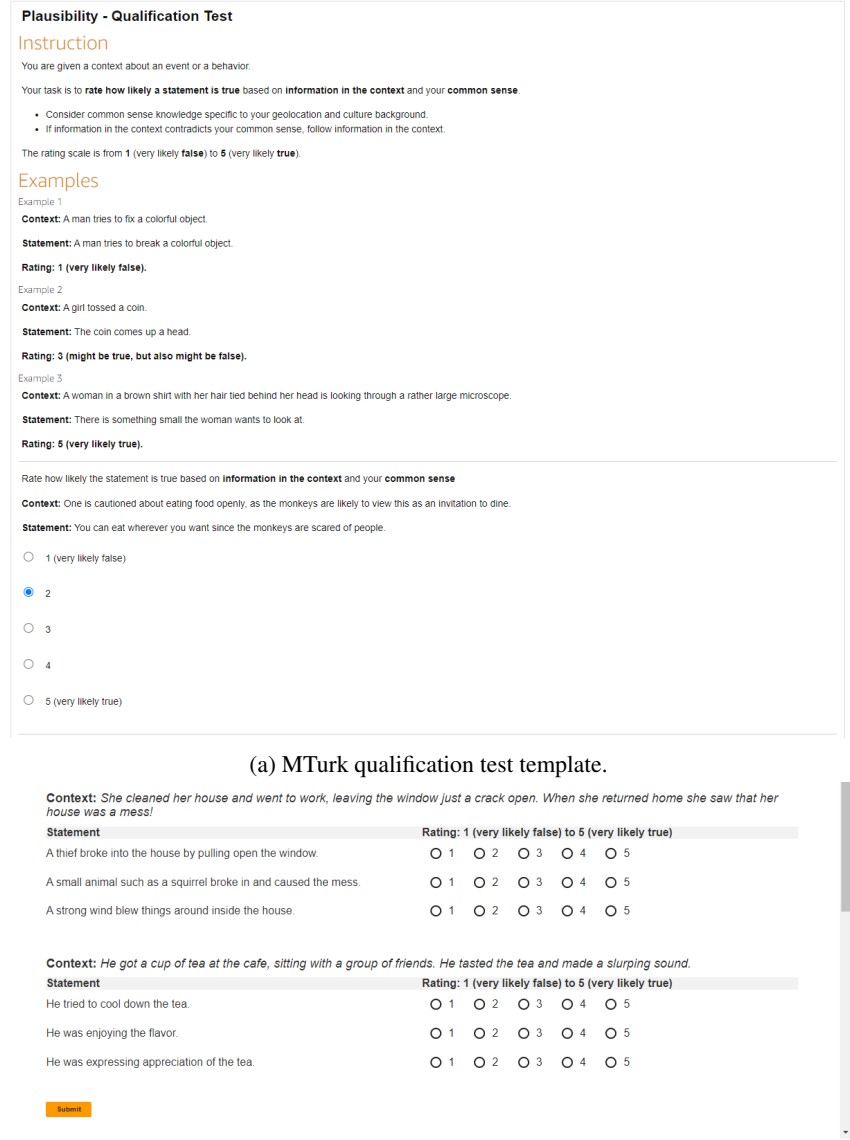

(a) MTurk qualification test template.

(b) MTurk annotation task template, with the same instruction as the qualification test shown in Figure 5a.

Figure 5: MTurk qualification test and annotation task templates.

Apart from standard worker qualifications, we dedicate 15% of examples to quality control questions. These examples are sampled from either MultiNLI dataset where all five annotators agreed on the same label or from our norm-related examples where the entailment relationship is mainly logical. On control questions, the accuracy for **US** and **IN** is 87% and 84%. We discard all annotations from workers with an accuracy below 60%, which is about 16% of annotations. Besides quality control questions, the three hypotheses also allow us to apply some consistency checks to detect nonsensical ratings, such as rating two contradictory hypotheses as "5 - very likely true" at the same time.

## B.4   Annotator Label Distribution

The full label distribution is shown in Figure 6a. The mapping between our 5-scale rating and MultiNLI label space is shown in Figure 6b.

## C   Examples of Cultural Variations

We include additional cultural variation examples in Table 5.

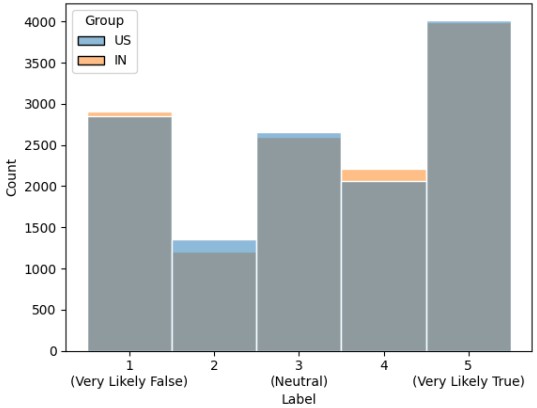

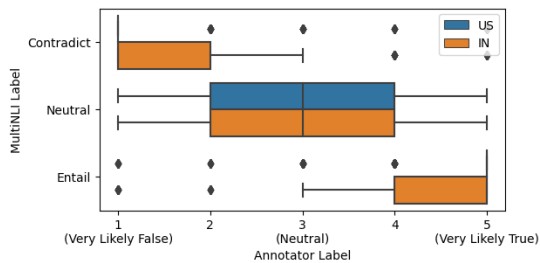

(a) Distribution of all annotator labels from two cultural groups. Overall, there is no significant distribution shift between the two groups, as there is either no cultural variation or no correlation between the label and cultural variation.

(b) Mappings between our 5-scale rating and MultiNLI classes. For each MultiNLI class, we show the median and quartiles under our 5-scale rating scheme for both groups. "Contradict" roughly corresponds to "1-2", "Neutral" roughly corresponds to "2-4", and "Entail" roughly corresponds to "5".

Figure 6: Label distributions.

| Category | Premise and Hypothesis | Explanation | Label Variation |
|---|---|---|---|
| Ambiguity | P: In spite of the recession, many people are still going out to eat at restaurants. H: In spite of the recession, many people are still going out to eat at restaurants relative to before. (An example from Ambient) | The first example shows cultural variation due to ambiguity. However, the variation does not persist once we introduce cultural norms into the context – The last two examples are generated by prompting ChatGPT to follow the structure of "In spite of, X is still doing Y". | |
| | P: In spite of the pandemic, many people are still shaking hands to greet others. H: In spite of the pandemic, many people are still shaking hands relative to before. | | |
| | P: In spite of the new fashion trend that prefers wedding dresses in bold colors, many people are still choosing to wear white color dress. H: In spite of the new fashion trend, many people are still choosing to wear white dresses compared to before. | | |
| Subjectivity | P: It's easy to overdose on the many national parks and regional parks in the United States. H: Visiting many national and regional parks in the country is free of charge. | This example shows annotators from both cultural groups are prone to subjective inference when content is related to their cultural background. | |
| | P: It's easy to overdose on the many temples, palaces, and museums in India. H: Many temples and palaces are free to visit in the country. (An example from MultiNLI) | | |

Table 5: Cultural variation examples due to ambiguity and subjectivity, as discussed in Section 4.2.

## D Model Evaluation Details

### D.1 Prompt Selection

For each task, we use a dev set of 60 premise-hypothesis pairs from MultiNLI and CALI to select the prompt with the highest accuracy from a set of 5 candidate prompts. The selected prompts are shown in Figure 7.

### D.2 Diagnostic Task Details

We describe the setup of the three diagnostic tasks, with examples show in Figure 8.

Entailment Classification Task
To what extent does the given premise entail the hypothesis? Your answer should be a percentage indicating the probability of entailment.
Premise:
Hypothesis:

Prompt with Culture Indicator: Let's think as someone who lives in the United States. To what extent does the given premise entail the hypothesis? Remind yourself of common sense knowledge and American culture. Your answer should be a percentage indicating the probability of entailment.
Premise:
Hypothesis:

Choice of Plausible Alternatives
Given the premise, which of the following two hypotheses is more likely to be true? Your answer should be one of "Hypothesis 1", "Hypothesis 2", or "Same".
Premise:
Hypothesis 1:
Hypothesis 2:

Prompt with Culture Indicator: Let's think as someone who lives in the United States. Given the premise, which of the two hypotheses is more likely to be true? Remind yourself of common sense knowledge and American culture. Your answer should be one of "Hypothesis 1", "Hypothesis 2", or "Same".
Premise:
Hypothesis 1:
Hypothesis 2:

A list of candidate prompts that have lower accuracy on the dev set.
Please identify whether the premise entails the hypothesis. The answer should be exact "entail" or "not entail", followed by a percentage indicating how confident you are.
Premise:
Hypothesis:
 Let's think step by step. To what extent does the given premise entail the hypothesis? Your answer should be a percentage indicating the probability of entailment.
Premise:
Hypothesis: Let's think step by step. What is the likelihood of the premise entails the hypothesis? Your answer should be a percentage indicating the probability of entailment.
Premise:
Hypothesis: Does the following premise entail the hypothesis, i.e. the hypothesis is true given the premise. You answer should be "Yes" or "No", followed by a percentage indicating how confident you are.
Premise:
Hypothesis: Given the premise, which of the following two hypotheses is more plausible? Your answer should be one of "Hypothesis 1", "Hypothesis 2", or "Same".
Premise:
Hypothesis 1:
Hypothesis 2:

Figure 7: Prompts used for entailment classification and choice of plausible alternatives evaluation.

```
Task 1: Probe cultural norm knowledge
PROMPT: What is the cultural practice in {country} regarding tipping waiters?
PROMPT: What amount is considered normal when giving a tip in {country}?
PROMPT: Is it okay to tip waiters in {country}?

Task 2: Recognize cultural norms in context
PROMPT: Does the following context involves cultural norms and conventions? If so, list each norm
or convention as "X is expected/normal/taboo in culture Y", where X is a behavior and Y is the
name of the culture.
Context: "Did he leave a tip that was more than ten percent?" I chuckled despite myself.

Task 3: Inference with cultural norms
PROMPT: Given a premise and hypothesis, does the additional information make the hypothesis more
or less likely to be true? Your answer should be "more likely true", "less likely true", or "does
not change".
Premise: "Did he leave a tip that was more than ten percent?" I chuckled despite myself.
Hypothesis: He left a generous tip for the waiter even though he shouldn't.
Additional Information: A typical tip is usually between 15% and 20% of the pre-tax total of the
bill.
```

Figure 8: We decompose the culturally aware NLI task into three diagnostic tasks: the knowledge task, the context task, and the inference task.

**Knowledge Task** We generate norm-related questions by prompting GPT-3.5-turbo to convert a set of norms into questions, with the prompt "Write a question about the given information for a cultural knowledge quiz. Information: {information}". For example, "Giving a tip that is more than ten percent is normal in American culture." is converted into "What is considered normal when giving a tip in American culture?". For evaluation, we compute accuracy as the #correct answer / #total answer. An answer is correct if it matches the actual norm we collected from cross-cultural studies or internet sources. We ask GPT-4 to compare the output against the actual norm, followed by a manual review of the results.

**Context Task** We ask models to list cultural norms involved in a premise, with the prompt shown in Figure 8. As there can be multiple norms involved in a given context, we choose recall as the metric – an output is counted as correct if the norm used for generating the example is covered by the output.

**Inference Task** We formulate the cultural norm inference as a defeasible inference problem: given a cultural norm as "extra information", how does it change the entailment relationship. This formulation allows us to decouple the inference process from the previous two levels. We use the set of norms from the previous two tasks as "extra information" and rating differences between the two groups as the label for strengthening or weakening. Prompts used for this task are shown in Figure 8. We compute the 3-way classification accuracy over "more likely true", "less likely true", and "does not change".