# OpenReview forum: "Culturally Aware Natural Language Inference"
_EMNLP/2023/Conference — EMNLP 2023 Findings_

### Official Review · Reviewer_PaX1 · 2023-08-01

**Typos Grammar Style And Presentation Improvements:** 003
**Soundness:** 3

**Excitement:**

3: Ambivalent: It has merits (e.g., it reports state-of-the-art results, the idea is nice), but there are key weaknesses (e.g., it describes incremental work), and it can significantly benefit from another round of revision. However, I won't object to accepting it if my co-reviewers champion it.

**Missing References:**

Looks good

**Paper Topic And Main Contributions:**

This paper addresses the problem that writers produce and readers consume
language in a particular cultural context, which includes culturally
specific knowledge about social norms.  This kind of knowledge is assumed
to be shared.  NLP systems are not always sensitive to this kind of
knowledge.  This paper introduces a culturally aware NLI dataset based on
English texts from the US and India, and uses it to investigate whether and
how current models are sensitive to cultural context.  The authors provide
an analysis of the dataset and of the performance of off-the-shelf systems
and zero-shot prompting approaches.

**Questions For The Authors:**

* A: Explain clearly what Table 2 shows us.  Specifically, what two numbers
  are you comparing (cite them) and what conclusions do you draw?  Consider
  my comments below.

* B: How do you think the LLM performances in Table 3 relate to the LLM
  performances in Table 2?

* C: Will you release the corpus?  How?

**Reasons To Accept:**

* The paper addresses an interesting issue in a novel way.

* The paper presents an interesting resource.

**Reasons To Reject:**

* Some important parts of the paper are hard to understand (Sections 3,
  5.1), despite mostly grammatical English and quite  abit of detail (maybe
  too much?).

* The categories developed in Section 4 are interesting but they are not
  used in a quantitative analysis.

* It is unclear what the takeaway from the evaluation of the LMs is, other
  than that GPTn is amazing.

* REMOVED. The use of the term "implicature" is confusing. [This is not a reason to reject after all.]

* The authors do not actually promise to release the resource.

**Reproducibility:**

3: Could reproduce the results with some difficulty. The settings of parameters are underspecified or subjectively determined; the training/evaluation data are not widely available.

**Reviewer Confidence:**

4: Quite sure. I tried to check the important points carefully. It's unlikely, though conceivable, that I missed something that should affect my ratings.

---

> ### Author Rebuttal · Authors · 2023-08-29
>
> ### Summary
> We thank the reviewer for their detailed feedback on experiment analysis and presentation. Following the suggestions, we performed additional analyses to help address the concerns below.
>
> ### Response per Question
>
> 1. *"The categories developed in Section 4 are interesting but they are not used in a quantitative analysis."*: Thanks for the suggestion! We add an analysis below to show how the categories developed in Section 4 can be used to perform error analysis of models.
>
> We use the semantic vs. pragmatic categorization to investigate which categories the model performs the best/worst. We evaluate on the same set of examples used in Table 2, which has about 12% semantic and 88% pragmatic cases. We focus our analysis on GPT-4 results in Table 2 (with the country indicator prompting).
>
> Example distribution per categories (semantic, pragmatic)
> | | Lexical Ambiguity (7%) | Referring Expressions (5%) | Violation of Maxims (9%) | Other Implicatures (79%) |
> |:---:|:---:|:---:|:---:|:---:|
> |GPT-4 US | 1.00 | 0.73 | 0.74 | 0.81|
> |GPT-4 IN | 1.00 | 0.61 | 0.82 | 0.72|
>
> We indeed find some interesting patterns: GPT-4 performs the best on the lexical ambiguity task and performs the worst on the Winograd schema style reference resolution task. For the reference resolution task, despite having the necessary cultural knowledge (as shown in the knowledge probing task), GPT-4 is less sensitive to cultural indicators presented in the prompt, but biases towards using syntactic cues.
>
> We show examples of GPT-4 predictions on the two categories below. The number is GPT-4 predicted probability of entailment.
>
> * Lexical Ambiguity
>   * P: I had some ladyfingers for dinner.
>   * H: I had some desserts for dinner.  US: 0.80  IN: 0.05
>   * H: I had some vegetables for dinner. US: 0.20 IN: 0.90
> * Referring Expressions
>   * P: She picked the white rose bouquet instead of the red rose bouquet, because it contrasts the color of traditional wedding attire.
>   * H: She picked the white rose bouquet instead of the red rose bouquet, because the white rose bouquet contrasts the color of traditional wedding attire. US: 1.00 IN: 1.00
>   * H: She picked the white rose bouquet instead of the red rose bouquet, because the red rose bouquet contrasts the color of traditional wedding attire. US: 0.00 IN: 0.00
>
> 2. *"takeaway from the evaluation of the LMs"*:
> (1) GPT-3.5/4 can achieve a higher accuracy on the US-label (culturally aware view) than All-label (culturally generic view), i.e. GPT-3.5 0.74 vs. 0.74 (+0), GPT-3.5+US 0.70 vs. 0.75 (+0.05), GPT-4 0.76 vs. 0.80 (+0.04), GPT-4+US 0.74 vs. 0.80 +0.06).
> (2) All models evaluated, including GPT-3.5/4, do not perform well on the IN label (a max F1 of 0.74 among all models for IN vs. a max F1 of 0.80 among all models for US), i.e. there is an accuracy gap between cultures.
>
> 3. *"How do you think the LLM performances in Table 3 relate to the LLM performances in Table 2?"*
> We discuss the connections in Section 5.2 and to recap:
>    * Table 3 serves as an error analysis for the GPT-3.5 and GPT-4 results in Table 2.
>    * We show that (1) higher accuracy on NLI tasks (Table 2 comparing GPT-3.5 vs. GPT-4: 0.74 vs. 0.80 on US and 0.72 vs. 0.75 on IN) corresponds to higher accuracy on all three diagnostic tasks, with the largest improvement observed on “context task” (Table 3 “all 40 behaviors” row: comparing 0.62 vs. 0.88). (2) Comparing across the three tasks in Table 3, the knowledge task is mostly saturated for the best model (GPT-4 0.98 on US and 0.93 on IN) context and inference tasks have larger room for improvement.
>    * The two observations suggest, apart from adding cultural knowledge to the model, we should also improve the model's ability to recognize when the cultural norm knowledge is relevant to the context and how to update inference with respect to normative behaviors.
>
>     Regarding the concern on whether these probing tasks are faithful explanations of the model performance on NLI task, we think this is an open question for prompting-based explanations (see Atanasova et al. 2023). However, if we define a model having certain knowledge as the model can pass at least one set of behavioral tasks that requires such knowledge, then we can make a claim about whether the model has certain knowledge, i.e. in our case, diagnosing whether a model knows a particular cultural norm or not.
>
> 4. *"The use of the term "implicature" is confusing."*
>     * In the context of "*culture-dependent inference is implicature by nature*" (and other places where we discuss inference): We use “Implicature” to refers to inferences that are cancellable, as opposed to textual entailment that are not (Line 203-205). Similar usage of “implicature” can be found throughout the NLI literature. See Manning, 2006, Gordon et al., 2012, Pavlick and Kwiatkowski, 2019, Rudinger et al., 2020 for a discussion on “implicature” vs. “textual entailment” and why “implicature” should be included in NLI (Line 151-154).
>
>        Culture-dependent inferences are implicatures, as culture-dependent inferences can always be canceled by adding an norm that counters the original one used in the inference. For example, “I didn't leave a tip.” implies “I am not happy with the service.” can be canceled by “because tipping is not practiced in this culture”.
>     * In the pragmatic dimension, i.e."*implicatures, including connotations of phrases*": We agree that the usage here is a bit confusing. We use it as a label to loosely refer to conversational implicatures. We will choose a better label to refer to this category.
>
> 5. *"The authors do not actually promise to release the resource."*: We are planning to release the corpus. Also see our response to ethics review (b).

---

### Official Review · Reviewer_QEVL · 2023-08-04

**Soundness:** 3

**Excitement:**

4: Strong: This paper deepens the understanding of some phenomenon or lowers the barriers to an existing research direction.

**Missing References:**

@inproceedings{liu-etal-2022-wanli,
    title = "{WANLI}: Worker and {AI} Collaboration for Natural Language Inference Dataset Creation",
    author = "Liu, Alisa  and
      Swayamdipta, Swabha  and
      Smith, Noah A.  and
      Choi, Yejin",
    booktitle = "Findings of the Association for Computational Linguistics: EMNLP 2022",
    month = dec,
    year = "2022",
    address = "Abu Dhabi, United Arab Emirates",
    publisher = "Association for Computational Linguistics",
    url = "https://aclanthology.org/2022.findings-emnlp.508",
    pages = "6826--6847"}

@inproceedings{chakrabarty-etal-2022-flute,
    title = "{FLUTE}: Figurative Language Understanding through Textual Explanations",
    author = "Chakrabarty, Tuhin  and
      Saakyan, Arkadiy  and
      Ghosh, Debanjan  and
      Muresan, Smaranda",
    booktitle = "Proceedings of the 2022 Conference on Empirical Methods in Natural Language Processing",
    month = dec,
    year = "2022",
    address = "Abu Dhabi, United Arab Emirates",
    publisher = "Association for Computational Linguistics",
    url = "https://aclanthology.org/2022.emnlp-main.481",
    pages = "7139--7159"}

**Paper Topic And Main Contributions:**

The paper proposes a new framework for cultural awareness in language understanding, consisting of knowledge of cultural norms, recognizing relevant norms for a context, and making appropriate culture-specific inferences. It introduces a new dataset CALI to evaluate this, with entailment judgments from two cultural groups, revealing differences due to cultural factors. The paper categorizes how norms influence understanding across lexical, semantic, pragmatic and script elements. Experiments show current models have norm knowledge but lack full awareness - they struggle to identify relevant norms for a context and adapt inferences. The paper advocates improving cultural awareness in models through task and model design. It also examines ambiguity and subjectivity as factors influencing cultural variations in entailment labels, providing a nuanced analysis of what drives differences.

**Questions For The Authors:**

See reasons to reject

**Reasons To Accept:**

1) It tackles an important problem of improving cultural awareness in NLP systems, which is becoming increasingly relevant as systems interact with diverse users globally. The framework and definitions around cultural awareness are useful contributions.
2) The new dataset CALI could be valuable - as it provides a concrete way to evaluate cultural awareness in inference tasks, something that has been lacking. Manually collecting premise-hypothesis pairs labeled by different cultural groups enables rigorous analysis.
3) The categorizations of how cultural norms influence language understanding is somewhat insightful. Considering lexical, semantic, pragmatic factors and links to behavioral scripts provides nuance.
4) The experiments on current LLMs highlight limitations in their cultural awareness at the knowledge vs context recognition vs inference levels. The analysis sheds light on where the gaps are.

**Reasons To Reject:**

1) The coverage of only two cultural groups (US and Indian) is extremely limited. The variations observed may not generalize to other cultures, reducing the usefulness of the dataset and analyses. While prior work has done this, such as CH-Wang et al. 2023 (https://arxiv.org/pdf/2305.14492.pdf) across English and Chinese, it is not particularly trying to create an NLI data set, but rather understanding differences in cultures via NLI and explanation. One of the limitations of this paper is that CALI is probably only relevant to study cultural NLI across two countries. I also feel like there are at least several works that have done similar things in the past on the same theme, such as one by Fung et al. (2022) (https://arxiv.org/pdf/2210.08604.pdf), who also basically treat <dialogue,socialnorm> as an NLI task, and the data is a lot bigger and more diverse. Therefore, I am unsure what is new here. Using a model in the loop technique to study cross-cultural differences, framing it as an NLI task, has already been done. This work seems heavily inspired, limiting at least some of the novelty aspects.

2) Relying heavily on language models like GPT-3 for data generation raises concerns about artifacts and bias in the data. The impact of model-produced biases is not sufficiently analyzed. We see some stuff in the Appendix, but this should be part of the main paper. The same holds for prompts. Too much of the results, and data depend on prompt sensitivity, rather than robust cultural reasoning. Additional experiments are needed to justify why a certain prompt is chosen and whether authors explored other prompting methodologies.

3) The majority vote for labels risks marginalizing minority cultural perspectives. Better aggregation methods should be explored. Authors should read and cite [https://direct.mit.edu/tacl/article/doi/10.1162/tacl_a_00449/109286/Dealing-with-Disagreements-Looking-Beyond-the]. Annotator disagreements may capture important nuances in subjective tasks like this, and their judgment is often ignored while aggregating annotations to a single ground truth. In order to address prior work, research has investigated the efficacy of multi-annotator models. This might be something to look into.

4) Biases related to gender, race, and other intersectional factors are not addressed sufficiently.

5) I am still not convinced why the authors are using sarcasm here. Is there any connection of the pragmatic insincerity of sarcasm comprehension with social norms? Additionally, prompting LLMs to generate narratives or opinions related to behavior, for example, “Describe a scene of tipping the waitress at a restaurant,” seems disconnected. Can you not use some situated, socially-grounded dialogue dataset that already exists?

6) I find the writing extremely hasty. The task is unclear. There needs to be a clear diagram that shows the entire pipeline from data creation to NLI. What is your task? What are the input, output, and label, etc.? The introduction of the Linguistic Dimension seems rather abrupt and ad hoc. Did the authors annotate for a linguistic dimension? If so, who is annotating these? MTurks cannot be qualified for such a task. Finally, why are the authors using r/AskEurope when their annotators are either from the US or India? Additionally, the US has lots of immigrants who might not be familiar with US-specific norms. Authors should have specific checks around whether someone has a significant presence in either country to be able to perform this task.

**Reproducibility:**

3: Could reproduce the results with some difficulty. The settings of parameters are underspecified or subjectively determined; the training/evaluation data are not widely available.

**Reviewer Confidence:**

4: Quite sure. I tried to check the important points carefully. It's unlikely, though conceivable, that I missed something that should affect my ratings.

---

> ### Author Rebuttal · Authors · 2023-08-29
>
> ### Summary
> We thank the reviewer for their feedback. We are encouraged that the reviewer found our dataset and analysis are valuable for improving cultural awareness in NLP. We also appreciate that the reviewer touched on many important open questions in the area, such as dataset coverage and biases, artifacts and biases of using LLMs, majority and minority cultural perspectives, reproducibility of GPT results which we address below.
>
> ### Response per Question
> 1. *Differences between our work and Fung et al.*: Our work and Fung et al. have very different goals. Fung et al and also CH-Wang et al, in their own term, aim to **discover norms** from existing corpus or online resources. The discovery process might use NLI models as an automated data processing tool. Our work revisits the NLI task itself, specifically **how cultural norms can lead to label variations in NLI**. We take this position since cultural factors have been largely ignored in the NLI literature, yet are critical for building culturally-aware NLP systems.
>
> 2. *"The coverage of only two cultural groups is extremely limited"*: We agree that having multiple cultural groups would be highly valuable. We will consider extending our dataset to cover more culture pairs in future work. For now, with two cultural groups, we can already observe how cultural variations contribute to NLI label variations. We also want to clarify that we do not expect the labels collected on a particular premise-hypothesis pair to generalize to other cultures, instead, we expect the proposed model-in-the-loop data collection framework to be reusable for studying NLI label variations in other cultures. We will emphasize this limitation of having only two cultural groups in our paper.
>
> 3. *"The impact of model-produced (GPT-3) biases is not sufficiently analyzed."* also *"Biases related to gender, race, and other intersectional factors are not addressed sufficiently."*: We will move the bias analysis in the appendix A.2 to the main text and discuss the impact on our dataset as follows: (1) Biases in data generation: We acknowledge that GPT-3 generated outputs are susceptible to artifacts and bias, in particular factual errors such as wrong culture-norm associations and stereotyped gender and social role associations (Lucy and Bamman, 2021; Bender et al., 2021; Yuan et al., 2021). As a mitigation measure, authors have manually inspected all premises and hypotheses generated by GPT-3 before sending for annotation and discard all examples that promote stereotyping. However, we want to caution users of the dataset that the removal process itself can still be affected by the subjectivity of the authors. (2) Biases in annotations: Culture can vary greatly within a locale due to social class, age, and gender. Thus, the annotations are best used for analyzing cultural factors in NLI. In particular, the annotations should not be assumed to generalize to all populations in the United States and India.
>
> 4. *"Additional experiments are needed to justify why a certain prompt is chosen and whether authors explored other prompting methodologies"*: We indeed conducted experiments on prompt selection. We construct a dev set for prompt selection "For each task, we use a dev set of 60 premise-hypothesis pairs from MultiNLI and CALI to select the prompt with highest accuracy from a set of 5 candidate prompts." (Appendix D.1). We also disclosed the full set of prompts that we have explored in the Appendix A Figure 4 and Appendix D Figure 9.
>
> 5. *"The majority vote for labels risks marginalizing minority cultural perspectives."*: We understand the concern here. In fact, we discuss the limitations of majority vote and mitigation (line 583-588). However, we also want to clarify labels in the dataset vs. labels used in modeling. Our dataset preserves the full label distribution and supports modeling techniques like multi-annotator. We also present the full label distribution whenever we show examples from the dataset (e.g. Table 1 and 4). The only two places we use the majority vote are when we compare with existing NLI dataset (Section 3.6) and when we evaluate existing NLI models (Section 5.1). We unfortunately have to use the majority vote in these cases for a fair comparison, as existing dataset, i.e. SNLI and MultiNLI, use majority vote as the label.
> We will include the paper suggested as a related work. Although the current work did not involve training new models, we think the modeling approach can be useful for future work that trains models on CALI.
>
> 6. *"Can you not use some situated, socially-grounded dialogue dataset that already exists?"*: We agree that “socially-grounded dialogue” is another interesting data source to explore. We view “narratives or opinions” and “socially-grounded dialogue” as **complementary** in studying how cultural factors lead to variations in language inference. Interpreting narratives and opinions allows us to measure direct reasoning about cultural norms. Narratives and opinions are also closer to the format of existing NLI datasets, i.e. SNLI and MultiNLI.
>
> 7. *"pragmatic insincerity of sarcasm comprehension"*: Why recognizing sarcasm involves social norms -- under Gricean pragmatics, sarcasm results from flouting; flouting involves violation of the Gricean maxims, such as quality (i.e. being truthful). Whether the listener perceives the speaker as being truthful can depend on the normative behavior of the situation.
>
> 8. *Presentation of the task and data creation pipeline*: We are happy to revise Figure 2 and task definition in Section 3.2 to make the task clear. To clarify here, Figure 2 shows the entire dataset creation pipeline, which include premises (task inputs), hypotheses (task inputs), and human annotations (i.e. task labels). Our task follows the standard NLI task, where input is a pair of premise and hypothesis, label is the entailment relationship between the premise and hypothesis. The difference is that we condition the label on the culture group of the listener (annotator), i.e. having more than one label per input. Concrete examples of the task are shown in Table 1.
>
> 9. *Presentation of linguistic dimension*: We (authors) annotated these dimensions ourselves by categorizing the label differences we observed on our dataset. We included additional results in response to review PaX1 on the cluster of examples we found per category and how we can use these categories for evaluation. Our additional results show that the linguistic dimension provides insights into which types of expressions models are good or bad at: For GPT-4, we found the model is perfect at resolving lexical ambiguity given culture indicators, yet struggles the most with referring expressions.
>
> 10. *Presentation of r/AskEurope*: The name r/AskEurope might be a bit misleading. r/AskEurope is not just about European cultural norms -- it is a community where people from different cultures (including immigrants and tourists) discuss the cultural differences between Europe and their own cultures, which include people from US and India. For example: norms around the structure of the name exist in many cultures https://www.reddit.com/r/AskEurope/comments/hapxks/whats_the_structure_of_names_in_your_country.
>
> 11. *"Specific checks around whether someone has a significant presence in either country"*: Yes, we did include a local-specific common sense question in the qualification test. About 50% of Amazon Turk workers passed the qualification test. (Appendix B.2 Annotation Quality Control).

---

### Official Review · Reviewer_61Gu · 2023-08-04

**Soundness:** 2

**Ethical Concerns:**

Yes

**Excitement:**

2: Mediocre: This paper makes marginal contributions (vs non-contemporaneous work), so I would rather not see it in the conference.

**Justification For Ethical Concerns:**

Given the nature of this work, it should be expected that an IRB review is present. Specifically, "Please indicate if your project was approved by an IRB (institutional review board)." in (https://2023.eacl.org/ethics/faq/)

**Paper Topic And Main Contributions:**

The paper gathers a series of cultural norm premises using prompting on GPT with online (i.e. reddit) and existing sources (normbank), further generates hypotheses of these premises (i.e. given a situation, what is the intention of the speaker? --> "the intention of the speaker is..."), and collects MTurk annotations from workers based in the US and those based in India on whether these hypotheses are likely to be true or false using a Likert scale; dataset size is 900 premises and 2.7K hypotheses. The authors then formalize this as a NLI task, mapping "very likely false" as "contradict", and "very likely true" as "entail".

The authors then test how different NLI models perform on this task in both culturally specific (US/India) and overall (majority vote from all) versions. In general, existing models are shown to perform pretty mediocre on all 3 settings, with GPT-4 & culturally-specific prompting performing slightly better than others.

**Questions For The Authors:**

How do annotator judgements vary across these two cultures? What does this say about these two societies? Any social insights that we can gain from this dataset?

**Reasons To Accept:**

The paper presents a novel dataset on cross-cultural judgments of normative behavior in a relatively unstudied setting in this area (US/India). The paper uses thorough human annotation, with each annotation labeled by 5 annotators. This dataset could prove useful for future studies in the area, towards computational studies of cultural norms across cultures.

**Reasons To Reject:**

The key concern I have with this paper is in evaluation. Given that the paper's framing and motivation are in "culturally aware" NLI, the paper does not go into, in any detail, what *cross-cultural insights* this dataset provides. Yes, the authors have collected a dataset of annotator judgments in different cultures. However, no analysis of how these norms may differ across cultures, in what qualitative or qualitative way, is given. Thus, it would seem to me that the key evaluation here should not be on how existing NLI models tuned on other NLI datasets would perform on this task, but rather on how these *cultures* fundamentally differ. Without this analysis, it presents several questions on whether such a dataset is ultimately useful. I would implore the authors to further evaluate their annotations in this way, to see if computational studies of cross-cultural norms provides any cultural insights.

EDIT: Thank you to the authors for providing more detail on what they see to be the specific contribution of this work. Nonetheless, my key gripe with this paper stands. To highlight that cultural factors contribute to label variations in NLI (and that they should indeed be considered) is, while entirely and accurately poised, not surprising—it would be hard to find researchers that disagree on the point that cultural factors and variations should be considered for many, if not all, traditional NLP tasks. Beyond this acknowledgment, it is precisely *how* we should take into account these cross-cultural differences that matters and informs future work; and on this front, the paper, in its current state, does not do a rigorous evaluation on the points listed above. Improving on these fronts would make this a much more rigorous paper.

**Reproducibility:**

5: Could easily reproduce the results.

**Reviewer Confidence:**

5: Positive that my evaluation is correct. I read the paper very carefully and I am very familiar with related work.

---

> ### Author Rebuttal · Authors · 2023-08-29
>
> ### Summary
> We would like to clarify the motivation and goal of this work.
>
> * Our goal is to highlight that **cultural factors contribute to label variations in NLI** and **cultural variations should be considered in language understanding tasks**.
>
> * Our work is largely motivated by two lines of research
>   * Social norms differ across cultures (To recap: Cultural differences have been noted in social behaviors and practices. See Geertz 1973; Triandis 1994; Hofstede et al., 2010. Recent work in NLP has computationally identified differences in social norms across cultures. See Yin et al., 2022; Nguyen et al., 2023; Fung et al., 2022; Ziems et al., 2023; CH Wang et al., 2023)
>   * Social norms are involved in language inference (To recap: Social norms are involved in reasoning about social judgment, intents and consequences, default assumptions, and ambiguity in expressions. See Forbes et al., 2020; Shen et al., 2022; Fung et al., 2022; Ziems et al., 2023, Emelin et al., 2021; Rashkin et al.; 2018, Abrams and Scheutz, 2022; Rudinger et al., 2020; Ziems et al.; Pyatkin et al., 2023)
>
>   These two lines of research suggest that people from different cultures likely make different inferences about social behaviors. Yet, language inference tasks such as NLI have been largely assumed to have a single label across cultures.
>
> Thus, our goal is **not** using the NLI task as a tool to study how norms differ across cultures. Instead, we revisit the assumptions and setup of the NLI task itself through a culture lens, showing cultural factors lead to label variations. With this goal in mind, we believe some concerns could be addressed.
>
> ### Response per Question
>
> 1. *"no analysis of how these norms may differ across cultures"*: Our work builds on previous work that performs extensive analysis on how certain norms differ across cultures, including work that studies tipping, dining, wedding customs, chronemics, waiting in line, politeness, etc [1-6], recent work in NLP that aims to curate knowledge bases of cultural norms (line 101-105), and online resources [7-8] (also line 239-246). In light of prior work, we focus our analysis on how cultural norms interact with language understanding, instead of cultural norm variations themselves. We choose NLI as a starting point, as it is one of the widely used and fundamental tasks in NLP in terms of language understanding.
>
> 2. *"key evaluation here should not be on how existing NLI models ... would perform on this task"*: Given the goal of this paper is to show cultural factors should be considered in NLI, it is natural and necessary to study how existing models that do not consider cultural factors fail on examples that involve cultural variations. Our results show that existing NLI models indeed do not transfer well to CALI examples involving cultural variations (Section 5.1).
>
> 3. *"cross-cultural insights"*, such as *“How do these cultures fundamentally differ? What does this say about these two societies? Any social insights that we can gain from this dataset?”*: While these are intriguing questions, existing cultural studies can already provide some insights. For example, Hofstede's Cultural Dimensions Theory shows that the U.S. and India differ significantly on Power Distance, Individualism, and Indulgence (Hofstede et al., 2010). Our work builds on top of these cultural analyses to provide insights into how cultural factors affect language understanding.
>
> 4. *"How do annotator judgements vary across these two cultures?"*: We provide a quantitative analysis in terms of statistical distribution of ratings (To recap: Annotators from the two cultural groups disagree on 30% of the labels; See Section 3.6 and Figure 7 in Appendix) and qualitative analysis on what types of inferences lead to variations in annotator judgment (To recap: We observe annotation disagreement on examples that involve lexical ambiguity, referring expressions, implicatures, and violations of Gricean maxims; See Section 4).
>
>
> References:
> 1. Lynn, M., & Lynn, A. (2004). National Values and Tipping Customs: A Replication and Extension. Journal of Hospitality & Tourism Research, 28(3), 356–364. https://doi.org/10.1177/1096348003256605
> 2. Rozin P. The meaning of food in our lives: a cross-cultural perspective on eating and well-being. J Nutr Educ Behav. 2005 Nov-Dec;37 Suppl 2:S107-12. doi: 10.1016/s1499-4046(06)60209-1. PMID: 16246277.
> 3. Buckley, E. (2006). A Cross-Cultural Study of Weddings through Media and Ritual: Analyzing Indian and North American Weddings. McNair Scholars Journal, 10, 3.
> 4. Macduff, I. (2006). Your Pace or Mine? Culture, Time, and Negotiation. Negotiation Journal, 22(1), 31–45. https://doi.org/10.1111/j.1571-9979.2006.00084.x
> 5. Fagundes, D. (2017), The Social Norms of Waiting in Line. Law and Social Inquiry, 42: 1179-1207. https://doi.org/10.1111/lsi.12256
> 6. Valentine, T.M. (1996). Politeness models in indian english.
> 7. Incluude. (2023). The Cultural Atlas. Retrieved from https://culturalatlas.sbs.com.au/
> 8. Wikipedia entries such as: https://en.wikipedia.org/wiki/Gratuity, https://en.wikipedia.org/wiki/Table_manners, https://en.wikipedia.org/wiki/Wedding_customs_by_country

---

### Meta-Review · Area_Chair_GSJ2 · 2023-09-23

**Recommendation:** 3

**Metareview:**

This paper presents a large dataset of NLI annotated with cultural awareness. The resulting CALI dataset would enable more research into how cultural differences affect the task of NLI. This work also opens future research directions with other languages and cultural differences. While, as pointed out by one reviewer, the dataset is limited to just one other culture of India, it is nevertheless challenging and important to build this cross-cultural dataset, and I encourage the authors (and our NLP research community) to broaden the set of cultures considered in future work.

There was some concern that this research collected human annotations but does not present IRB approval. Please describe how the research ethics were considered and adhered when collecting the annotations and go through IRB if such process exists in your research institution.

---

### Decision · Program_Chairs · 2023-10-07

**Decision:**

Accept-Findings

**Comment:**

This paper presents a large dataset of NLI annotated with cultural awareness. The resulting CALI dataset would enable more research into how cultural differences affect the task of NLI. This work also opens future research directions with other languages and cultural differences. While, as pointed out by one reviewer, the dataset is limited to just one other culture of India, it is nevertheless challenging and important to build this cross-cultural dataset, and I encourage the authors (and our NLP research community) to broaden the set of cultures considered in future work.

There was some concern that this research collected human annotations but does not present IRB approval. Please describe how the research ethics were considered and adhered when collecting the annotations and go through IRB if such process exists in your research institution.